# In Silico Molecular Studies of Antiophidic Properties of the Amazonian Tree *Cordia nodosa* Lam.

**DOI:** 10.3390/molecules24224160

**Published:** 2019-11-16

**Authors:** Carmen X. Luzuriaga-Quichimbo, José Blanco-Salas, Luz María Muñoz-Centeno, Rafael Peláez, Carlos E. Cerón-Martínez, Trinidad Ruiz-Téllez

**Affiliations:** 1Faculty of Health Sciences Eugenio Espejo, University UTE, Quito 170147, Ecuador; luzuriaga.cx@gmail.com; 2Department, Faculty of Sciences, University of Extremadura, 06006 Badajoz, Spain; truiz@unex.es; 3Department of Botany, University of Salamanca, 37008 Salamanca, Spain; luzma@usal.es; 4Departament of Pharmaceutical Sciences, Organic Chemistry, University of Salamanca, 37008 Salamanca, Spain; pelaez@usal.es; 5Herbarioum Alfredo Paredes, QAP, Ecuador Central University UCE, Quito 170521, Ecuador; carlosceron57@hotmail.com

**Keywords:** *Cordia*, in silico, antiophidic, quercetrin, docking, validation

## Abstract

We carried out surveys on the use of *Cordia nodosa* Lam. in the jungles of Bobonaza (Ecuador). We documented this knowledge to prevent its loss under the Framework of the Convention on Biological Diversity and the Nagoya Protocol. We conducted bibliographic research and identified quercetrin as a significant bioactive molecule. We studied its in silico biological activity. The selected methodology was virtual docking experiments with the proteins responsible for the venomous action of snakes. The molecular structures of quercetrin and 21 selected toxins underwent corresponding tests with SwissDock and Chimera software. The results point to support its antiophidic use. They show reasonable geometries and a binding free energy of −7 to −10.03 kcal/mol. The most favorable values were obtained for the venom of the Asian snake *Naja atra* (5Z2G, −10.03 kcal/mol). Good results were also obtained from the venom of the Latin American *Bothrops pirajai* (3CYL, −9.71 kcal/mol) and that of Ecuadorian *Bothrops asper* snakes (5TFV, −9.47 kcal/mol) and *Bothrops atrox* (5TS5, −9.49 kcal/mol). In the 5Z2G and 5TS5 L-amino acid oxidases, quercetrin binds in a pocket adjacent to the FAD cofactor, while in the myotoxic homologues of PLA2, 3CYL and 5TFV, it joins in the hydrophobic channel formed when oligomerizing, in the first one similar to α-tocopherol. This study presents a case demonstration of the potential of bioinformatic tools in the validation process of ethnobotanical phytopharmaceuticals and how in silico methods are becoming increasingly useful for sustainable drug discovery.

## 1. Introduction

*Cordia* is a tropical genus of arbustive *Boraginaceae* and is quite interesting from a pharmacological point of view [1]. More than thirty species are referenced as medicinal [2], having bioactive compounds such as rosmarinic acid, cordiaquinones and cordiachromes [3,4]. *Cordia alliodora* Cham., one of the most important timber trees in the Amazon, has an interesting chemical profile [5,6,7,8,9,10,11] with antimicrobial, antifungal, larvicidal [12] and cytotoxic activities [13] which have been experimentally tested. *Cordia verbenaceae* DC has been studied as anti-inflammatory [14], analgesic [15], antibacterial [16], antiallergic [17] and antitumoral [18]; the latter activity is attributed to the rosmarinic acid (Figure 1) [19].

Another *Cordia* with promising properties is *Cordia nodosa* Lam, (= *Cordia collococa* Aubl [20]) a Pan-Amazonian species that contains [13] quercetrin (Figure 2), a strong antiproliferative in vitro.

In Amazonian Ecuador, many ethnic groups (cofan, redwood, siona, wao, shuar, achuar and kichwa) have reported references to the ancestral use [21]. The fruit is edible, the wood is employed for the construction of their houses in the jungle, and cultural ceremonies and rites are prepared with the leaves [21]. *Cordia nodosa* Lam contains phenols that justify its anti-inflammatory and analgesic applications and its moderate bactericidal action [22]. However, the most interesting ancestral knowledge of this plant is its ability to act as an antidote to snake bites [23]. This problem, seldom considered in Europe, affects millions of inhabitants of tropical areas of the planet and has not developed a pharmaceutical research according to its dimension [24]. Every year, about 5.4 million snake bites produce 1.8–2.7 million cases of poisoning, 81,410–137,880 deaths and about three times as many amputations and other permanent disabilities [24]. The World Health Organization has included snake bites in the category of “Neglected and Forgotten Diseases” [24].

In the context of ethnobotanical research conducted by our team in the Ecuadorian Amazon [21], we carried out surveys on the use of *C. nodosa* under the Framework of the Convention on Biological Diversity and the Nagoya Protocol [25]. We had two specific aims: (a) to describe the current use as an antivenom of *C. nodosa* in the Bobonaza Basin (Pastaza, Ecuador) and (b) to offer an in silico validation by searching the scientific literature and by overall performing docking tests.

## 2. Results

### 2.1. Ethnobotanical Survey

The medicinal uses given to the species retrieved from our fieldwork prospections and literature review are summarized in Table 1, which shows that the use of *C. nodosa* as an antiophidic is currently in force in indigenous Amazonian ethnic groups. Names like kuchamanku, awas, putunmuyu, machakuymisunsal or machakuykaspi have been published previously, but not paluwapu (“= snake stick”), which we learned from the canelo-kichwas cultures of Pastaza that we worked with [21]. We found that when a snake bite occurs, they take the bark, cook it for about one minute, and then drink the resulting liquid in a single dose.

### 2.2. Chemical and Activity Prospection: Results of the Bibliographic Review

The main component of the extract was quercetrin, a 3α-l-rhamnoside of quercetin. The genine has a chemical structure based on a C6-C3-C6 carbon skeleton, with a chromene ring bearing a second aromatic ring at position 2. Therefore, it is a flavonoid, specifically a flavonol (Figure 2). This is a chemical group in which antiophidic properties are known [28].

The literature review performed is summarized in the following tables. It is known that snake venom comprises peptides and proteins that act as cytotoxins, neurotoxins, hemotoxins or myotoxins. The 21 molecular targets of snakebite poisonings, retrieved from our bibliographic research, are shown in Table 2.

### 2.3. Docking

The liaison energies of quercetrin with the studied targets are presented in Table 3. They have been colored by groups according to their similarity to the rest of the sequences (see Appendix A for details). They oscillate between −10.03 kcal/mol and −7.01 kcal/mol.

Figure 3, Figure 4, Figure 5, Figure 6, Figure 7, Figure 8, Figure 9, Figure 10, Figure 11, Figure 12, Figure 13, Figure 14 and Figure 15 show the molecular models of the quercetrin binding with the targets of Table 3 (left) and the corresponding 2D interaction diagram generated with LigPlot + (right) [50], made with UCSF Chimera Software. Toxins are represented in golden yellow, quercetrin in blue, and original ligands in pink. The small squares (**a**) show the toxin-quercetrin complex in the most favorable arrangement. When it occupies the hollow of another ligand present in the structure of the target, it has been preserved (shown as thin sticks in pink) to allow a comparison. The augmented figures (**b**) show, in detail, the dispositions of the quercetrin molecule between the chains of the toxins.

## 3. Discussion

The traditional use of this plant for its antiophidic properties has been previously documented by the Shuar and the Napo-Rune people of neighboring provinces, although the method of application is different. In these cases, the plant is applied directly to the affected place, or they chew the young leaves and the fruits. They also prepare infusions, not with the bark, but with the root or the juice of the stem [23]. Published references on the activity of quercetrin have indicated that it inhibits lipoxygenase svPLA2 [51] and hyaluronidase NNH1 [52], neutralizing the hemorrhagic venom of *Bothrops jararaca* [51,52,53], a Latin American snake.

In the docking tests that we carried out, the toxic snakes studied showed very high affinities with quercetrin. There were formed complexes of energy comparable to the ones with original targets such as 4GEW or 5A4W. Detailed information about these protein homologies has been included in Appendix B (Table A1). Thus, reasonable binding free energy values of −7 to −10 kcal/mol were obtained. The most favorable values were for the venom of the Asian snake (Chinese cobra or Taiwan cobra) *Naja atra* (5Z2G, −10.03 kcal/mol) and the Latin American *Bothrops pirajai* (3CYL, −9.71 kcal/mol). Very good results were also found with the 5TFV of the Ecuadorian snakes *Bothrops asper*, (ΔG −9.47 kcal/mol) and 5TS5 of *Bothrops atrox* (ΔG −9.49 kcal/mol). Therefore, quercetrin can not only be used as an antiophid for Ecuadorian venomous snakes, but for many others.

The action can be expected to be effective, especially because, in addition, the models have not presented unfavorable interactions according to SwissDock scoring terms. On many occasions, the quercetrin molecule is placed in pockets that occupy other known ligands of the targets used in this study. This is the case for 3CYL and 3CXI (Figure 5b and Figure 6b), which occupy the pocket of α-tocopherol, 5TS5 (Figure 4b), which is close to that of FAD, 6CE2 (Figure 7b), that occupies that of suramin (a known inhibitor), 1XXS (Figure 9b), that of two stearic acids, 1QLL (Figure 13b), that of tridecanoic acid, and 2W12 (Figure 15b), which occupies the peptidomimetic inhibitor site. All of this reinforces the validity of the results of the performed docking tests.

In the 5Z2G and 5TS5 L-amino acid oxidases, quercetrin binds in a pocket adjacent to the FAD cofactor, while in the myotoxic homologues of PLA2, 3CYL and 5TFV, it joins in the hydrophobic channel formed when oligomerizing in the first one, similar to α-tocopherol.

These facts reinforce the validity of the traditional use reported. They will have to be corroborated in vitro, in vivo, and even with subsequent clinical trials. Nevertheless, this is encouraging evidence in the field of finding new solutions to this pathology.

## 4. Materials and Methods

### 4.1. Ethnobotanical Survey

All the information referring to the ethnobotanical study from which the data derives is available in Appendix B, which contains references to voucher specimens, authorizations and permissions. Table 1 summarizes the medicinal uses of the species retrieved from our fieldwork prospections and literature data.

### 4.2. Chemical and Activity Prospection: Bibliographic Review

A bibliographic review was carried out following the PRISMA Group method [54]. The databases accessed were Academic Search Complete, Agricola, Agris, Biosis, CABS, Cochrane, Cybertesis, Dialnet, Directory of Open Access Journals, Embase, Espacenet, Google Patents, Google Academics, Medline, PubMed, Science Direct, Scopus, Teseo, and ISI Web of Science. The selected citations were summarized, and a critical reading allowed us to develop the discussion.

### 4.3. Docking

The molecular docking method applied comprises the following procedures: ligand preparation, protein selection, docking, and analysis of the results. The energies produced after docking, interaction residues and interaction types were studied for the analyses following general procedure for molecular docking [55,56]. Docking was performed with the SwissDock Docking Web Service (Available online: http://www.loc.gov). A 3D quercetrin virtual structure was built with Spartan^®^, Wavefunction Inc. Selection of targets was made based on a bibliographic review of natural bioactive compounds against snake bites [28,54]. A total of 21 venoms from snakes (targets) with known X-ray structures were tested (see Table 2, Table 3 and Appendix A). Molecular structures were consulted in the Protein Data Bank (PDB), and the reference IDs were taken to include them in the Swiss Dock Program. The target + ligand set was considered stable when the values of the binding free energy were lower than −7 kcal/mol. This consideration is based on docking experiments with the known X-ray structures 4GUE and 5A4W complexes of quercetrin resulting in binding energies values of −9.30 and −8.28 kcal/mol, respectively. Once the results of the docking were obtained, they were analyzed with UCSF Chimera.

## 5. Conclusions

The information obtained from the ethnobotanical investigations carried out in the Bobonaza Basin (Ecuador) allowed us to verify the good capacity in silico of quercetrin, the active ingredient obtained from *Cordia nodosa*. The binding energies of quercetrin with all the macromolecules (toxins from venoms of different snakes) were adequate, since they were all less than −7 kcal/mol.

The in silico docking evaluation combined with ethnobotanical information was very effective as a research method. It allowed us to select the appropriate active principle from the beginning, thus avoiding the tedious previous work of testing principle assets that have no references and therefore working blindly with molecules that would not couple to these toxins. The search for new bio-products oriented from basic ethnobotanical knowledge is an investigation that could result in products with great therapeutic efficacies.

## Figures and Tables

**Figure 1 molecules-24-04160-f001:**
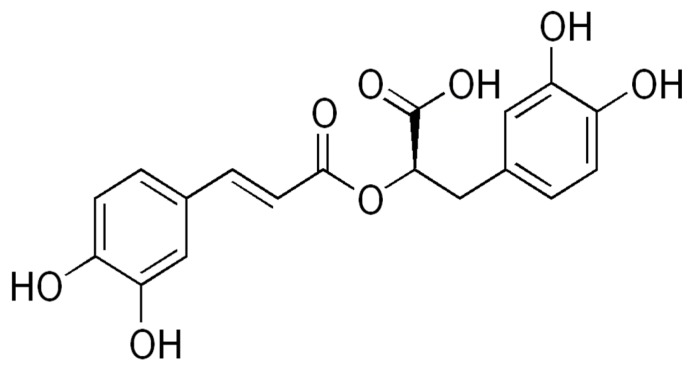
Rosmarinic acid.

**Figure 2 molecules-24-04160-f002:**
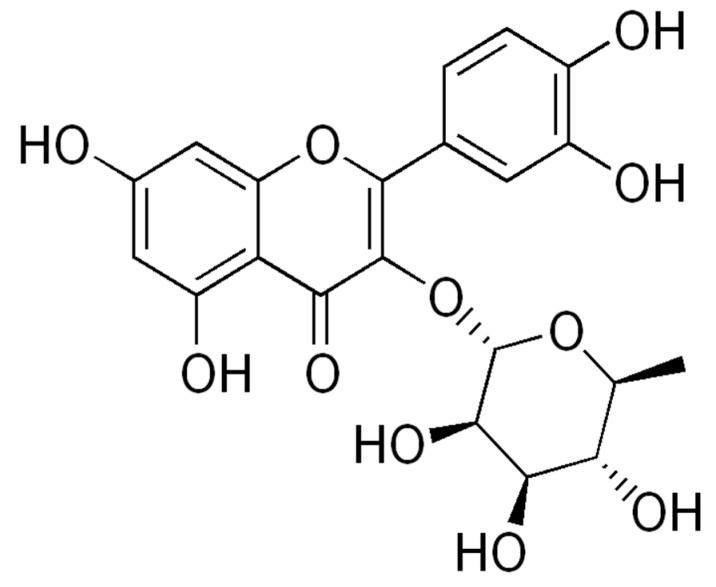
Quercetrin.

**Figure 3 molecules-24-04160-f003:**
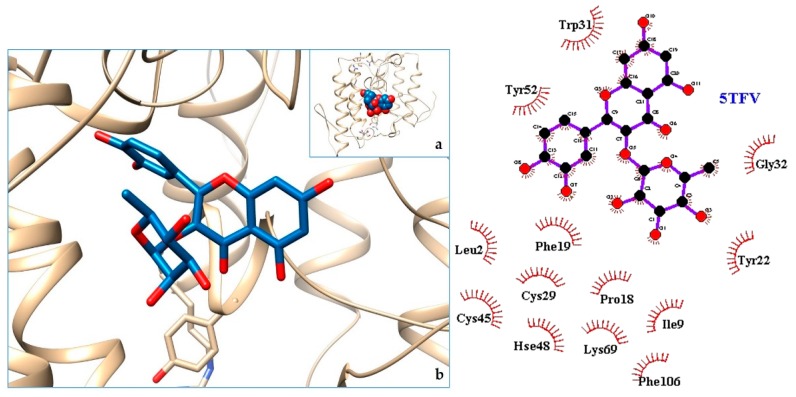
5TFV-quercetrin complex. in the most favourable arrangement (**a**) and augmented (**b**) showing the disposition between the chain of toxins. Right: 2D interaction diagram.

**Figure 4 molecules-24-04160-f004:**
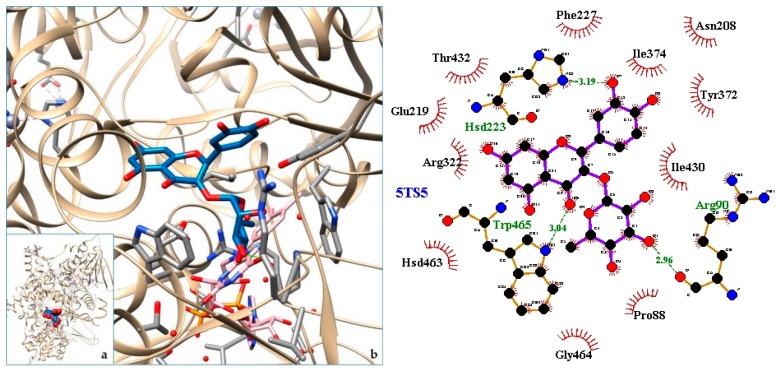
5TS5-quercetrin complex. in the most favourable arrangement (**a**) and augmented (**b**) showing the disposition between the chain of toxins. Right: 2D interaction diagram.

**Figure 5 molecules-24-04160-f005:**
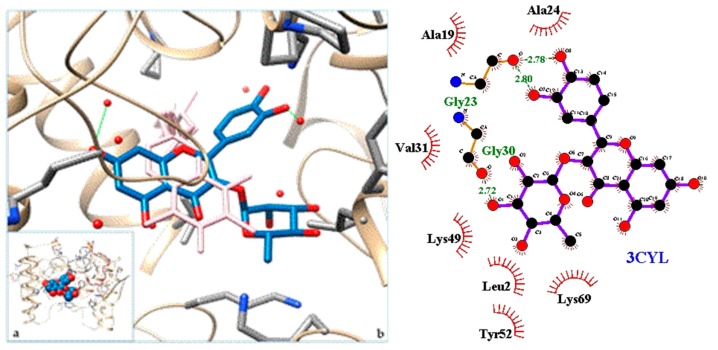
3CYL quercetrin complex in the most favourable arrangement (**a**) and augmented (**b**) showing the disposition between the chain of toxins. Right: 2D interaction diagram.

**Figure 6 molecules-24-04160-f006:**
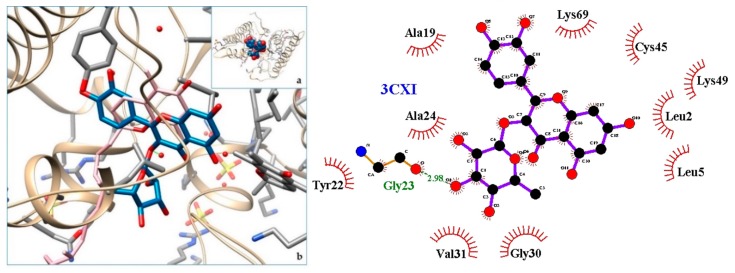
3CXI-quercetrin complex in the most favourable arrangement (**a**) and augmented (**b**) showing the disposition between the chain of toxins. Right: 2D interaction diagram.

**Figure 7 molecules-24-04160-f007:**
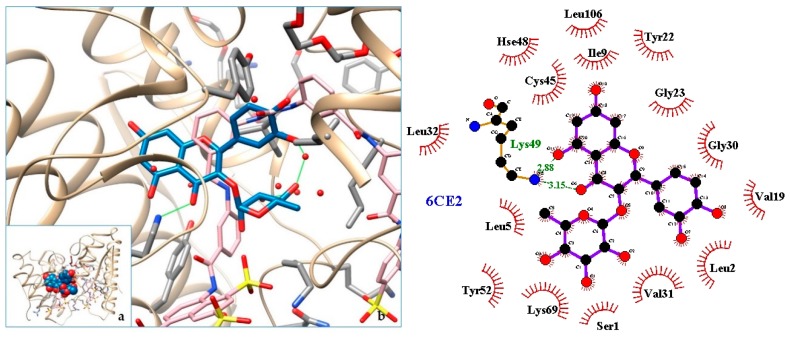
6CE2-q uercetrin complex in the most favourable arrangement (**a**) and augmented (**b**) showing the disposition between the chain of toxins. Right: 2D interaction diagram.

**Figure 8 molecules-24-04160-f008:**
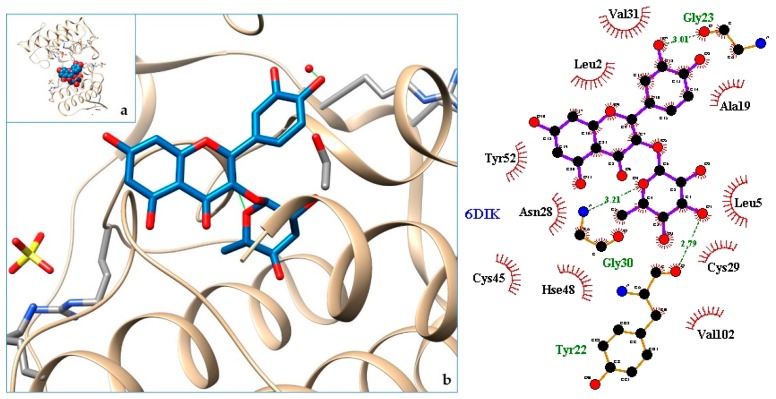
6DIK-quercetrin complex in the most favourable arrangement (**a**) and augmented (**b**) showing the disposition between the chain of toxins. Right: 2D interaction diagram.

**Figure 9 molecules-24-04160-f009:**
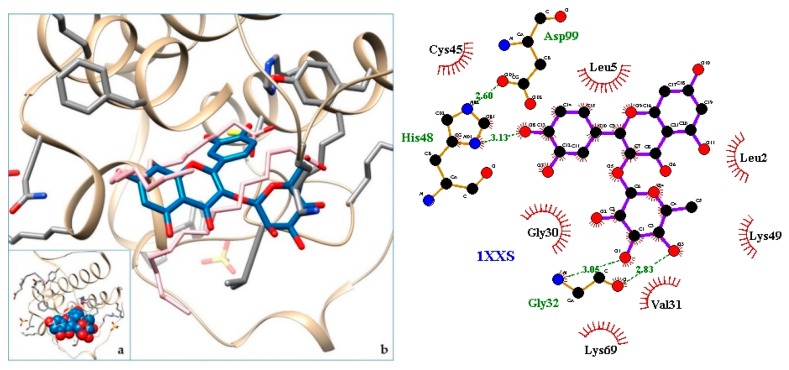
1XXS-quercetrin complex in the most favourable arrangement (**a**) and augmented (**b**) showing the disposition between the chain of toxins. Right: 2D interaction diagram.

**Figure 10 molecules-24-04160-f010:**
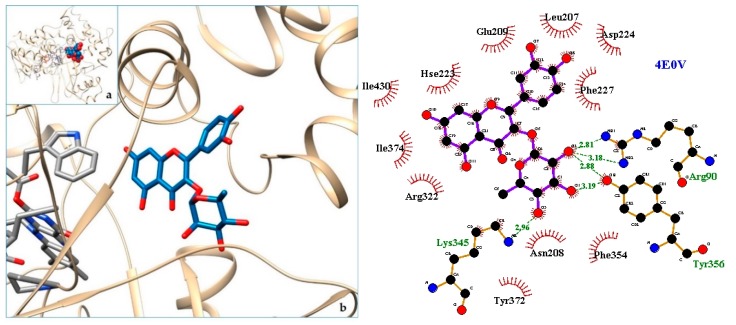
4E0V-quercetrin complex in the most favourable arrangement (**a**) and augmented (**b**) showing the disposition between the chain of toxins. Right: 2D interaction diagram.

**Figure 11 molecules-24-04160-f011:**
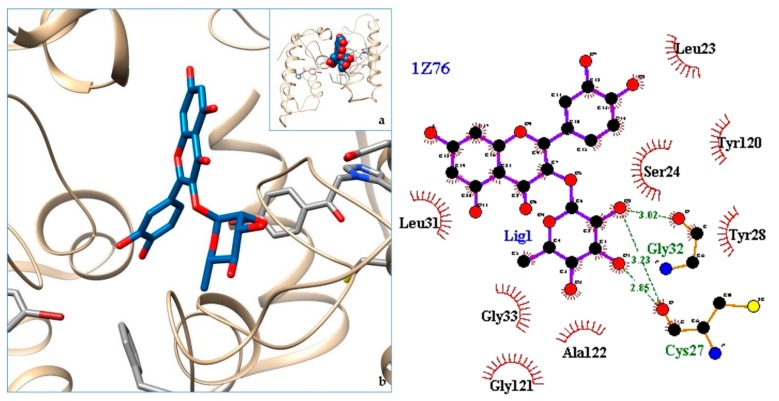
1Z76-quercetrin complex in the most favourable arrangement (**a**) and augmented (**b**) showing the disposition between the chain of toxins. Right: 2D interaction diagram.

**Figure 12 molecules-24-04160-f012:**
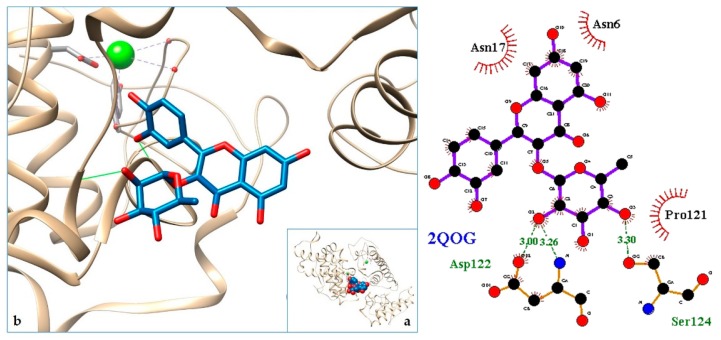
2QOG-quercetrin complex in the most favourable arrangement (**a**) and augmented (**b**) showing the disposition between the chain of toxins. Right: 2D interaction diagram.

**Figure 13 molecules-24-04160-f013:**
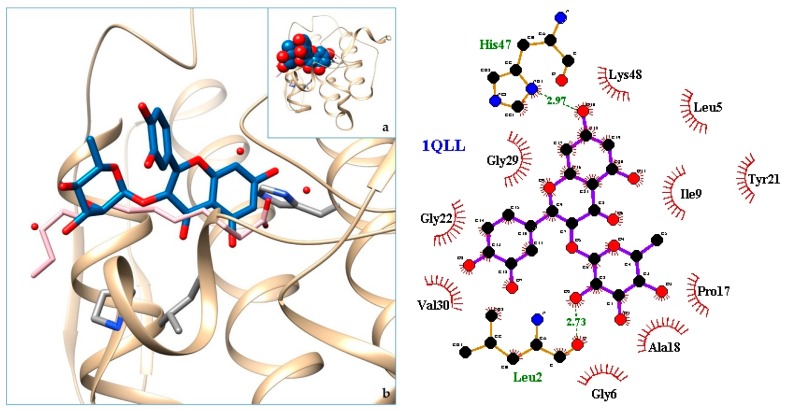
1QLL-quercetrin complex in the most favourable arrangement (**a**) and augmented (**b**) showing the disposition between the chain of toxins. Right: 2D interaction diagram.

**Figure 14 molecules-24-04160-f014:**
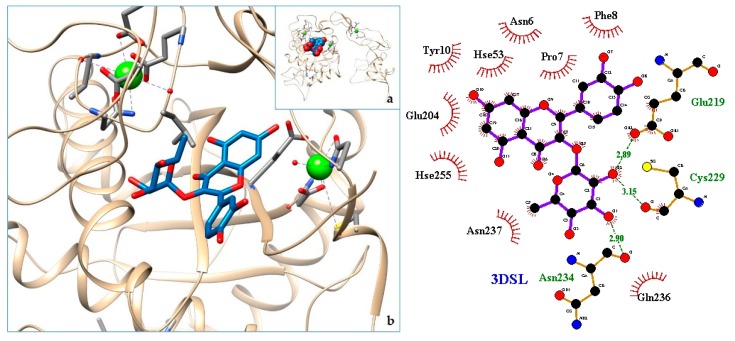
3DSL-quercetrin complex in the most favourable arrangement (**a**) and augmented (**b**) showing the disposition between the chain of toxins. Right: 2D interaction diagram.

**Figure 15 molecules-24-04160-f015:**
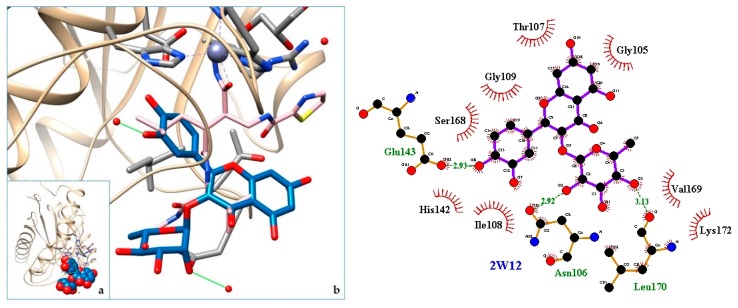
2W12-quercetrin complex in the most favourable arrangement (**a**) and augmented (**b**) showing the disposition between the chain of toxins. Right: 2D interaction diagram.

**Table 1 molecules-24-04160-t001:** Medicinal uses given to the species retrieved from our fieldwork prospections and literature review.

Organ/System	Part Used	Formulation	Traditional Knowledge	Ethnic Group-Province (Country)	Reference
Circulatory system	leaves	decoction	hypertension	Amerindian NorthWest District (Guiana)	[26]
Digestive system			gases	Siona-Sucumbíos (Ecuador)	[23]
Respiratory system	bark	cooking	treat cough	Secoya-Sucumbíos (Ecuador)	[23]
	stem				
	inner bark	finely grate and decoction	cold and shortness of breath	Amerindian (French Guiana)	[26]
	leaves	decoction	whooping cough	Amerindian NorthWest District (Guiana)	[26]
	fruit	suck	snot in babies	Amerindian NorthWest District (Guiana)	[26]
Musculature and skeleton	leaves	crush the leaves and rub the body with them	rheumatism, sprains, muscle aches, bruises	Amerindian NorthWest District (Guiana)	[26]
Nervous system and mental illness	leaves	baths with the decoction of the leaves	madness and psychiatric disorders	Yanesha (Perú)	[27]
		decoction	headache	Amerindian NorthWest District (Guiana)	[26]
Symptoms and states of undefined origin	bark		indeterminate conditions	Secoya-Sucumbíos (Ecuador)	[23]
	flowers			Kichwa del Oriente-Orellana (Ecuador)	
	fruit		energizing	Wao-Orellana (Ecuador)	
	leaves	infusion	dizziness	ethnicity not specified-Napo (Ecuador)	
		decoction	fever	Amerindian NorthWest District (Guiana)	[26]
Poisoning	leaves	apply directly in the affected place	spider bite, to decrease inflammation and prevent gangrene	East Kichwa-Napo and Orellana (Ecuador)	[23]
	fruit				
	plant	cooking			
	bark	cooking			
	root	cooking			
	bark	scraped and in water	snake bites, to decrease inflammation and prevent gangrene	East Kichwa, Shuar-Napo, Orellana, Pastaza, Sucumbíos (Ecuador), Piaroa (Venezuela)	[23,26]
		infusion			
	root	infusion			
	stem	juice			
	fruit	juice			
	young leaves	chewed			
	leaves	apply directly in the affected place			

**Table 2 molecules-24-04160-t002:** Toxins from Ecuadorian (1–2), Latin American (3–13) or non-American (14–21) snakes, and the corresponding Protein Data Base Identifier (PDB ID).

Toxin	PDB ID	Reference
1. MT-I—Basic phospholipase a2 myotoxin iii	5TFV	[29]
2. LAAO—L-amino acid oxidase from *Bothrops atrox*	5TS5	[30]
3. PLA2—Phospholipase A2: Piratoxin I (myotoxic Lys49-PLA2) from *Bothrops pirajai*	3CYL	[31]
4. PLA2—Phospholipase A2: BthTX-I—Bothropstoxin I from *Bothrops jararacussu* venom/	3CXI	[31]
5. PLA2—Phospholipase A2: Myotoxin (MjTX-I) from *Bothrops moojeni*	6CE2	[32]
6. PLA2—Phospholipase A2: Bothropstoxin I (BthTX-I)	6DIK	[33]
7. svPLA2—Phospholipase A2: myotoxin II from *Bothrops moojeni*	1XXS	[34]
8. LAAO—L-amino acid oxidasefrom the *B. jararacussu* venom	4E0V	[35]
9. svPLA2—Acidic phospholipase A2 (BthA-I) from *Bothrops jararacussu*	1Z76	[36]
10. VRV-PL-V—Crotoxin B, the basic PLA2 from *Crotalus durissusterrificus*	2QOG	[37]
11. PLA2—Piratoxin-II (Prtx-II) - a K49 PLA2 from *Bothrops pirajai*	1QLL	[38]
12. Bothropasin, the Main Hemorrhagic Factor from *Bothrops jararaca* venom	3DSL	[39]
13. SVMP—P-I snake venom metalloproteinase BaP1	2W12	[40]
14. NNH1—L-amino acid oxidase from venom of *Naja atra*	5Z2G	[41]
15. LAAO—L-amino acid oxidase from *Vipera ammodytes* venom	3KVE	[42]
16. PDE I—Phosphodiesterase (PDE) from Taiwan cobra (*Naja atra atra*) venom	5GZ4	[43]
17. VRV-PL-V—Phospholipase ACII4 from Australian King Brown Snake (*Pseudechis australis*)	3V9M	[44]
18. NN-PL-I—Phospholipase A2 from indian cobra (*Naja naja*)	1PSH	[45]
19. LAAO—L-amino acid oxidase from Agkistrodon Halys Pallas (*Gloydius halys*)	1REO	[46]
20. NNH1—Phosphodiesterase (PDE) fromTaiwan cobra (*Naja atra atra*)	5GZ5	[47]
21. PLA2—Phospholipase A2 (Pla2) from *Naja naja*	1A3D	[48,49]

**Table 3 molecules-24-04160-t003:** The liaison energies of quercetrin with the PDB ID studied targets.

1. 5TFV	−9.71	MT-I—Basic Phospholipase a2 Myotoxin iii
2. 5TS5	−9.47	LAAO—L-amino acid oxidase from *Bothrops atrox*
3. 3CYL	−9.49	PLA2—Phospholipase A2: Piratoxin I (myotoxic Lys49-PLA2) from *Bothrops pirajai*
4. 3CXI	−9.37	PLA2—Phospholipase A2: BthTX-I—Bothropstoxin I from *Bothrops jararacussu* venom/
5. 4GUE	−9.30	N-terminal kinase domain of RSK2 with flavonoid glycoside quercetrin
6. 6CE2	−9.19	PLA2—Phospholipase A2: Myotoxin (MjTX-I) from *Bothrops moojeni*
7. 6DIK	−9.16	PLA2—Phospholipase A2: Bothropstoxin I (BthTX-I)
8. 1XXS	−9.01	svPLA2—Phospholipase A2: myotoxin II from *Bothrops moojeni*
9. 4E0V	−8.96	LAAO—L-amino acid oxidasefrom the *B. jararacussu* venom
10. 1Z76	−8.56	svPLA2—Acidic phospholipase A2 (BthA-I) from *Bothrops jararacussu*
11. 2QOG	−8.32	VRV-PL-V—Crotoxin B, the basic PLA2 from *Crotalus durissusterrificus*
12. 5A4W	−8.28	AtGSTF2 from *Arabidopsis thaliana*
13. 1QLL	−8.23	PLA2—Piratoxin-II (Prtx-II) - a K49 PLA2 from *Bothrops pirajai*
14. 3DSL	−8.20	Bothropasin, the Main Hemorrhagic Factor from *Bothrops jararaca* venom.
15. 2W12	−7.71	SVMP—P-I snake venom metalloproteinase BaP1

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
