# Peer review of "In Silico Molecular Studies of Antiophidic Properties of the Amazonian Tree Cordia nodosa Lam."

_molecules, 2019, doi:10.3390/molecules24224160_

Round 1

Reviewer 1 Report

The paper has described the in silico molecular studies of quercetrin, which is a bioactive compound containing in Cordia nodosa Lam. C. nodosa Lam is traditionally used as anti-inflammatory and anti-analgesic agents, but of particular interest is used as antidotes to snakebite, which is often a severe issue in tropical areas and is included in "Neglected and Forgotten Diseases" in the WHO. In this paper, the authors bioinfomatically analysized interaction of quercetrin against proteins responsible for venome and compared binding mode of quercetrin with each protein. The results may be useful as bioinfomatic tools for in silico drug discovery, and the paper is acceptable for publication in Molecules. However, the following points should be considered for the revision of this paper to be accepted.

(1) The authors should discuss protein homology such as structure, sequence, binding site, consensus region and so on, in comparison with each protein containing 4GEW and 5A4W.

(2) The authors should explain the rationale for why targets 1-13 as modeling structures were chosen.

Minor points,

"Quercetrin" >> "quercetrin," not capitalized except beginning of the sentence.

p. 1, Line 3 from the bottom: "DC." >> "DC" remove the period.

p. 2, Line 14-16: "Every year…permanent disabilities" >> References are needed.

p. 2, Line 20: "Nagoya Protocol." >> References are needed.

p. 3, Table 1: Regarding "(NW Guiana)," ethic group-provinces are needed in front of parentheses. And the abbreviation of NW is needed in the table legend.

p. 5 Figure 3-5: The order of docking results should be matched to that in Table 2.

p. 9, Line 13: "4 GEW" >> "4GEW"

p. 9, Line12-13: "There were…5A4W" >> references are needed.

p. 10, Line 2: 4.1. "Etnobotanical" >> 4.1. "Ethnobotanical"

p. 10, Line 15: Reference "[54]" lacks in the reference section.

p. 10, Line 20: "ray" >> "X-ray".

Reference section: The authors should check reference style such as page number, book, web sites, and so on, and follow the journal style.

Reference 14: not an appropriate citation.

Author Response

The paper has described the in silico molecular studies of quercetrin, which is a bioactive compound containing in Cordia nodosa Lam. C. nodosa Lam is traditionally used as anti-inflammatory and anti-analgesic agents, but of particular interest is used as antidotes to snakebite, which is often a severe issue in tropical areas and is included in "Neglected and Forgotten Diseases" in the WHO. In this paper, the authors bioinfomatically analysized interaction of quercetrin against proteins responsible for venome and compared binding mode of quercetrin with each protein. The results may be useful as bioinfomatic tools for in silico drug discovery, and the paper is acceptable for publication in Molecules. However, the following points should be considered for the revision of this paper to be accepted.

(1) The authors should discuss protein homology such as structure, sequence, binding site, consensus region and so on, in comparison with each protein containing 4GEW and 5A4W.

We have calculated the sequence identity and similarity matrix for all the sequences. The results are shown as a heat map in the following table: <25% yellow, 25% -50% orange, 50-75% green and >75% light blue. According to these results the proteins have been grouped in several clusters (indicated by different colours in the upper row of the table. We have added a table to the supplementary material and a comment on the discussion of the selected protein targets. 4GEW and 5A4W show little similarity to each other and to the rest of the studied proteins.

4GUE

5A4W

1QLL

1XXS

1Z76

2QOG

2W12

3CXI

3CYL

3DSL

4E0V

5TFV

5TS5

6CE2

6DIK

4GUE

5A4W

1QLL

1XXS

1Z76

2QOG

2W12

3CXI

3CYL

3DSL

4E0V

5TVF

5TS5

6CE2

6DIK

The similarity clustering grouped proteins with similar functions and ligand binding functions, as shown in the following modified table 3 and legend

1.                 5TFV

−9.71

MT-I—Basic phospholipase a2 myotoxin iii

[29]

2.                 5TS5

−9.47

LAAO—L-amino acid oxidase from Bothrops atrox

[30]

3.                 3CYL

−9.49

PLA2—Phospholipase A2: Piratoxin I (myotoxic Lys49-PLA2) from Bothrops pirajai

[31]

4.                 3CXI

−9.37

PLA2—Phospholipase A2: BthTX-I—Bothropstoxin I from Bothrops jararacussu venom/

[31]

5.                 4GUE

−9.30

N-terminal kinase domain of RSK2 with flavonoid glycoside quercitrin

6.                 6CE2

−9.19

PLA2—Phospholipase A2: Myotoxin (MjTX-I) from Bothrops moojeni

[32]

7.                 6DIK

−9.16

PLA2—Phospholipase A2: Bothropstoxin I (BthTX-I)

[33]

8.                 1XXS

−9.01

svPLA2—Phospholipase A2: myotoxin II from Bothrops moojeni

[34]

9.                 4E0V

−8.96

LAAO—L-amino acid oxidasefrom the B. jararacussu venom

[35]

10.              1Z76

−8.56

svPLA2—Acidic phospholipase A2 (BthA-I) from Bothrops jararacussu

[36]

11.              2QOG

−8.32

VRV-PL-V—Crotoxin B, the basic PLA2 from Crotalus durissusterrificus

[37]

12.              5A4W

−8.28

AtGSTF2 from Arabidopsis thaliana

13.              1QLL

−8.23

PLA2—Piratoxin-II (Prtx-II) - a K49 PLA2 from Bothrops pirajai

[38]

14.              3DSL

−8.20

Bothropasin, the Main Hemorrhagic Factor from Bothrops jararaca venom.

[39]

15.              2W12

−7.71

SVMP—P-I snake venom metalloproteinase BaP1

[40]

Table 3. The liaison energies of Quercetrin with the 1-21 PDB ID studied targets. Similar targets according to their sequence are equally coloured.

(2) The authors should explain the rationale for why targets 1-13 as modeling structures were chosen.

Targets 1-13 were selected for the docking studies as they are known snake venoms from the literature whose X-ray structures have been published. The docking section of the materials and methods has been reworded in order to clarify this point.

Minor points,

"Quercetrin" >> "quercetrin," not capitalized except beginning of the sentence. Ok, done.

1, Line 3 from the bottom: "DC." >> "DC" remove the period. Ok, done.

2, Line 14-16: "Every year…permanent disabilities" >> References are needed. Ok, done

2, Line 20: "Nagoya Protocol." >> References are needed. Ok, done

3, Table 1: Regarding "(NW Guiana)," ethic group-provinces are needed in front of parentheses. And the abbreviation of NW is needed in the table legend. . Ok, done

5 Figure 3-5: The order of docking results should be matched to that in Table 2. Ok, done

9, Line 13: "4 GEW" >> "4GEW" Ok, changed

9, Line12-13: "There were…5A4W" >> references are needed. Ok, included.

10, Line 2: 4.1. "Etnobotanical" >> 4.1. "Ethnobotanical" Ok, changed.

10, Line 15: Reference "[54]" lacks in the reference section. Ok, it was a mistake. Deleted

10, Line 20: "ray" >> "X-ray". Ok, changed

Reference section: The authors should check reference style such as page number, book, web sites, and so on, and follow the journal style. Ok,  we have used Mendeley Tool to improve this topic. Afterwords we have reviewed it for final mistakes.

Reference 14: not an appropriate citation. Ok,  changed

Thank you for your time and consideration of our work.

Best regards,

José Blanco Salas

Reviewer 2 Report

This is an interesting study. I enjoyed the informative writing with a tone of field trip summary. I would though suggest some more calculations to be added for making the computational modeling part useful for further use.

Docking is an empirical calculation and as such binding energy terms are meaningless. Docking is never meant to be guiding tool to find molecule(s) that will not bind a given receptor, hence it locates binding poses (false positives) more easily than finding a non-binder. The authors need to compare their docking protocol against compounds that are known to bound venom proteins. A comparative energetics will then make more meaning to do such calculations. In the same line, they should do docking with molecules that do not have anti-venom properties, and do not bind to the target proteins, for the sake of completeness. Are these docking calculations site directed or blind docking? Are there well defined binding sites present in all the protein targets to bind ligand? Arey they conserved? Structurally similar? Stability of ligand binding can simply be validated by running routine molecular dynamics simulations. This will really point whether the docking simulations pointing to false negatives. There are several free MD engines available to run such calculations. 

Author Response

Reviewer 2

This is an interesting study. I enjoyed the informative writing with a tone of field trip summary. I would though suggest some more calculations to be added for making the computational modeling part useful for further use.

Docking is an empirical calculation and as such binding energy terms are meaningless. Docking is never meant to be guiding tool to find molecule(s) that will not bind a given receptor, hence it locates binding poses (false positives) more easily than finding a non-binder. The authors need to compare their docking protocol against compounds that are known to bound venom proteins. A comparative energetics will then make more meaning to do such calculations. In the same line, they should do docking with molecules that do not have anti-venom properties, and do not bind to the target proteins, for the sake of completeness.

We agree with the reviewer on the fact that empirical energies are by themselves meaningless and can only be used in a comparative context. In this regard, the results here presented have been compared to the docking energies obtained applying the same protocol to two published X-ray structures of quercitrin complexes with two targets (4GUE and 5A4V). The docking section of the materials and methods has been rewritten in order to better explain this fact. The results presented here are therefore analysed in comparison with the values obtained for these reference structures. The blind docking protocol successfully reproduced the experimentally determined structures and therefore the docking energies can be considered as bona fide comparison values for binders.

With respect to the docking of molecules without antivenom properties, it is difficult to propose decoys with enough similarity to quercitrin that have “proven” no anti-venom activity. Similar results to the ones found here would similarly need future experimental confirmation and would therefore not be really informative. We have therefore opted for a comparison with itself in a true positive target.

Are these docking calculations site directed or blind docking?

All the results presented are blind docking studies. The finding of usual binding pockets of the targets to bind supports the performance of the docking strategy used. Furthermore, the docking protocol applied has been shown to correctly predict the binding poses of ligands in their targets in blind docking experiments. These results imply the comparison of the energies of binding for the same ligand to different binding pockets, which therefore imply different interacting amino-acids and degrees of burial, as would occur in proteins with no sequence similarity or different binding pockets.

Are there well defined binding sites present in all the protein targets to bind ligand? Are they conserved? Structurally similar?

As indicated in figures 3 – 15 and their legend, quercitrin in many cases binds at or close to the binding pockets of other ligands present in the X-ray structures. When this happens, the original ligand has been conserved in the figure in thin wireframes in pink (e.g. in Fig. 3 quercitrin binds in the site of vitamin E of 3CYL and in figure 5 it binds next and partially overlapping with FAD of 5TS5) in order to account for this fact. The similarity of the targets has been now included in the analysis (see reviewer 1). We have also added a 2D plot of the molecular interactions and aminoacids involved in quercitrin binding (modified figures 3-15) for each protein.

Stability of ligand binding can simply be validated by running routine molecular dynamics simulations. This will really point whether the docking simulations pointing to false negatives. There are several free MD engines available to run such calculations. 

Molecular dynamics studies are indeed a good approach to discard false positives. We have applied them to find false negatives as well by applying them prior to docking studies in binding sites that had experienced size reductions due to induced fit. In the actual context they could be used to discard false negatives by opening closed cavities in potential targets. However, they require long simulation times and preparation beyond the scope of this work aimed at a fast determination of likely binders from natural sources based on ethnobotanical applications.

Thank you for your time and consideration of our work.

Best regards,

José Blanco Salas

Reviewer 3 Report

1. The present study is interesting and it provide some scientific information for the development of the products with great therapeutic activities. However, more information about the interactive sites between the receptors and ligands, for example the interactive amino acids, the bonding type such as electrostatic, ionic bonds, and so on, which is very important for us to understand the results of the present study.

2. There are several references related to the present study which have been listed as followings. The authors were suggested to read or compared them with the present study, especially the experimental design, the analysis of results and so on.

[1] Tu, Maolin; Cheng, Shuzhen; Weihong; et al (2018). Advancement and prospects of bioinformatics analysis for studying bioactive peptides from food-derived protein: sequence, structure, and functions[J]. TrAC Trends in Analytical Chemistry, 105, 7-17. DOI: 10.1016/j.trac.2018.04.005

[2] Tu, Maolin; Wang, Cong; Chen, Cheng; et al (2018). Identification of a novel ACE-inhibitory peptide from casein and evaluation of the inhibitory mechanisms[J]. Food Chemistry, 256, 98-104. DOI: 10.1016/j.foodchem. 2018.02.107

Author Response

The present study is intresting and it provide some scientific information for the development of the products with great therapeutic activities. However, more information about the interactive sites between the receptors and ligands, for example the interactive amino acids, the bonding type such as electrostatic, ionic bonds, and so on, which is very important for us to understand the results of the present study.

We have added for each complex a 2D plot of the molecular interactions and aminoacids involved (modified figures 3-15), generated with LigPlot+.

There are several references related to the present study which have been listed as followings. The authors were suggested to read or compared them with the present study, especially the experimental design, the analysis of results and so on.

[1] Tu, Maolin; Cheng, Shuzhen; Weihong; et al (2018). Advancement and prospects of bioinformatics analysis for studying bioactive peptides from food-derived protein: sequence, structure, and functions[J]. TrAC Trends in Analytical Chemistry, 105, 7-17. DOI: 10.1016/j.trac.2018.04.005

[2] Tu, Maolin; Wang, Cong; Chen, Cheng; et al (2018). Identification of a novel ACE-inhibitory peptide from casein and evaluation of the inhibitory mechanisms[J]. Food Chemistry, 256, 98-104. DOI: 10.1016/j.foodchem. 2018.02.107

Thank you very much for these two references. We have included in the revised version.

Thank you for your time and consideration of our work.

Best regards,

José Blanco Salas

Round 2

Reviewer 1 Report

The revised paper has been improved. However, 2D interaction diagram in Figure 3-15 should be modified to be clear (character and number of amino acid). After being revised, this paper is recommended to be published.

Author Response

The revised paper has been improved. However, 2D interaction diagram in Figure 3-15 should be modified to be clear (character and number of amino acid). After being revised, this paper is recommended to be published.

We have redrawn figures 4-16. Aminoacid numbers and distances are now much easier to read.

Thank you for your time and consideration of our work.

Best regards,

José Blanco Salas

Reviewer 2 Report

Thanks for providing a revised version. There are still few points left that need to be addressed. There are some typos as well as statements that require expanding the central idea.

"Figures 3–15 show the molecular models of each of the quercetrin couplings with targets 1–13 of Table 32 (left) ant the" Replace the "ant"  

Please replace "coupling"s with binding.   

"The action can be expected effective, especially because, in addition, the models have not presented unfavourable interactions." What unfavorable interaction the authors talking about? Did you check the ligand strain energy related to bound and unbound forms?

What are those references in Table 3? Are those for the corresponding PDB files? If so do mention them as a table footnote. Else it is confusing.

There are a lot of figures 3-15 showing bound complexes. What did they convey to a reader? A reader is not going to look in to them and generate conclusions from them. Please explain the need/idea behind putting them in the main text. If they are for graphical representations only, put them in the suppliments. IS there a common interaction pattern? 

Author Response

Thanks for providing a revised version. There are still few points left that need to be addressed. There are some typos as well as statements that require expanding the central idea.

"Figures 3–15 show the molecular models of each of the quercetrin couplings with targets 1–13 of Table 32 (left) ant the" Replace the "ant"   Ok Done

Please replace "coupling"s with binding.   Ok Done

"The action can be expected effective, especially because, in addition, the models have not presented unfavourable interactions." What unfavorable interaction the authors talking about? Did you check the ligand strain energy related to bound and unbound forms?

The statement is based upon visual inspection of the ligand contacts and the scoring of the docking software, which gives scores for different interactions, including intra and intermolecular. A brief explanation is now given in the text.

What are those references in Table 3? Are those for the corresponding PDB files? If so do mention them as a table footnote. Else it is confusing.

References in table 3 correspond to the original publications of the PDB files. They have been removed from table 3 as they appear already in table 2.

There are a lot of figures 3-15 showing bound complexes. What did they convey to a reader? A reader is not going to look in to them and generate conclusions from them. Please explain the need/idea behind putting them in the main text. If they are for graphical representations only, put them in the suppliments.

We have included the figures showing the bound complexes because they provide an overall view of the drug – target interaction and binding site location, including a comparison with other ligands binding to those sites. The details have been included as 2D diagrams in order to make the contacts more explicit after a reviewer requirement. They could help to answer to questions such as the one following. 

IS there a common interaction pattern?

There is no common binding pattern. When proteins o high similarity are compared, the ligand binds in many case in the same pocket, but in different poses. This also partially reflects the different side chain dispositions of the sidechains. However, for similar proteins there is a recognition of a common binding site.

Thank you for your time and consideration of our work.

Best regards,

José Blanco Salas

Reviewer 3 Report

After revision, the present study was described better and the quality was improved well. It was recommended to be published after language checking.

Author Response

After revision, the present study was described better and the quality was improved well. It was recommended to be published after language checking.

Thank you very much, but this paper was translated by MDPI Traslation Services (English_Editing_MDPI_english-12746).

Thank you for your time and consideration of our work.

Best regards,

José Blanco Salas
